# Aspirin Foliar Spray-Induced Changes in Light Energy Use Efficiency, Chloroplast Ultrastructure, and ROS Generation in Tomato

**DOI:** 10.3390/ijms26031368

**Published:** 2025-02-06

**Authors:** Julietta Moustaka, Ilektra Sperdouli, Emmanuel Panteris, Ioannis-Dimosthenis S. Adamakis, Michael Moustakas

**Affiliations:** 1Department of Botany, Aristotle University of Thessaloniki, 54124 Thessaloniki, Greeceepanter@bio.auth.gr (E.P.); 2Institute of Plant Breeding and Genetic Resources, Hellenic Agricultural Organisation-Demeter (ELGO-Demeter), 57001 Thessaloniki, Greece; esperdouli@elgo.gr; 3Section of Botany, Department of Biology, National and Kapodistrian University of Athens, 15784 Athens, Greece; iadamaki@biol.uoa.gr

**Keywords:** biostimulant, photosystem II (PSII), chlorophyll content, chlorophyll fluorescence, photoprotection, excess excitation energy, electron transport, photorespiration, peroxisomes

## Abstract

Aspirin (Asp) is extensively used in human health as an anti-inflammatory, antipyretic, and anti-thrombotic drug. In this study, we investigated if the foliar application of Asp on tomato plants has comparable beneficial effects on photosynthetic function to that of salicylic acid (SA), with which it shares similar physiological characteristics. We assessed the consequences of foliar Asp-spray on the photosystem II (PSII) efficiency of tomato plants, and we estimated the reactive oxygen species (ROS) generation and the chloroplast ultrastructural changes. Asp acted as an osmoregulator by increasing tomato leaf water content and offering antioxidant protection. This protection kept the redox state of plastoquinone (PQ) pull (q*p*) more oxidized, increasing the fraction of open PSII reaction centers and enhancing PSII photochemistry (Φ*_PSII_*). In addition, Asp foliar spray decreased reactive oxygen species (ROS) formation, decreasing the excess excitation energy on PSII. This resulted in a lower singlet oxygen (^1^O_2_) generation and a lower quantum yield for heat dissipation (Φ*_NPQ_*), indicating the photoprotective effect provided by Asp, especially under excess light illumination. Simultaneously, we observed a decrease in stomatal opening by Asp, which reduced the transpiration. Chloroplast ultrastructural data revealed that Asp, by offering a photoprotective effect, decreased the need for the photorespiration process, which reduces photosynthetic performance. It is concluded that Asp shares similar physiological characteristics with SA, having an equivalent beneficial impact to SA by acting as a biostimulant of the photosynthetic function for an enhanced crop yield.

## 1. Introduction

Aspirin (Asp), the trade name for acetylsalicylic acid (ASA), is extensively used as an anti-inflammatory non-steroidal drug and as an analgesic, antipyretic, and anti-thrombotic therapeutic agent for cardiovascular prophylaxis [1]. It has potential applications in cancer treatment, but it is used extensively also as an antibacterial agent [2]. Additionally, due to its multiple molecular targets, it is used to relieve the symptoms of COVID-19 [3]. Salicylic acid (SA), which is produced naturally in plant chloroplasts and its concentration increases in response to biotic or abiotic stresses, can be produced by the hydrolysis of ASA [4].

SA is a fundamental plant hormone which has garnered considerable attention due to its beneficial function as an antioxidant and plant growth regulator under biotic and abiotic stresses [5,6,7,8]. SA impacts a variety of physiological, biochemical, and developmental processes, e.g., stomatal closure, membrane permeability, seed germination, photosynthesis, transpiration rate, plant immunity, and growth yield [4,9,10,11]. At minimal concentrations, SA controls physiological processes, mainly during abiotic stress conditions [12,13,14,15,16,17]. Synthesis of osmolytes, like proline, is induced by SA under salt stress [18], drought stress [19,20] or heavy metal stress [21]. However, the regulation of proline metabolism by SA differs in different plants and diverse stresses [17]. SA impact on plants may vary depending on the plant species, the concentration, the exposure time, and the application [4,22]. For example, SA foliar spray increased photosynthetic function of corn and soybean under greenhouse conditions [23,24], but decreased net photosynthesis of maize plants under normal growth conditions [25]. However, the same concentration provided protection against low-temperature stress in maize plants [25]. Yet, SA alleviated the harmful effects of paraquat [26], and diminished water deficit effects in radish [27].

SA application, by decreasing chlorophyll content in tomato plants, reduced phototoxicity and provided photoprotection in photosystem II (PSII), improving PSII function [4]. Increasing the photochemical efficiency of the absorbed light energy improves photosynthesis and plant productivity [28,29], serving as a faster alternative to genetic engineering for enhancing crop yields [30]. The method of chlorophyll *a* fluorescence has been widely employed to explore the function of the photosynthetic machinery and especially of PSII [31,32,33,34,35], providing evidence about the light energy use efficiency [36,37]. The light energy absorbed by PSII antennae can either be utilized for photochemistry or dissipated via other various regulated or non-regulated processes [38,39].

Despite the chemical and physiological characteristic similarities between ASA and SA, the former has been used only in a few biological experiments [40,41,42,43,44,45,46,47]. ASA has been shown to have either positive or negative effects on seed germination and various growth parameters depending on the concentration used [43]. Exogenous applications of ASA were reported to reduce cold stress effects on common beans (*Phaseolus vulgaris* L.) by increasing total soluble sugars and proline accumulation [45], and to induce multiple stress tolerance in bean and tomato plants [41]. Moreover, ASA has been shown to provide plant resistance to disease and pests and to boost the plant’s immune system [40].

ASA increased foliage weight per plant in sweet potato genotypes, while there were no influences on root yield [44]. Also, no phytotoxicity was observed by immersion of potato tubers in ASA solutions [42]. Conversely, ASA has been considered a potential environmental pollutant with drug toxicity effects [48], and many alerts have been raised about the ecological damage of Asp abuse [49]. When ASA was introduced into the soil, negative effects were observed on both the crustacean *Heterocypris incongruent* and on spring barley [46]. ASA coupled with hypoxia exposure caused hepatic tissue damage in an estuarine fish (*Mugilogobius chulae*), by affecting energy metabolism, antioxidant regulation, and autophagy processes [47].

Tomato (*Solanum lycopersicum* L.) is, after potato, the most important vegetable, being recognized for its high diet and health rank with a high economic value [50,51,52]. Due to its importance as a nutrition source, extensive investigation has been carried out to enhance tomato yield and improve its fruit quality [53]. In the present study, we hypothesize that, since Asp and SA share physiological characteristic similarities, foliar spray of tomato plants with Asp, will result in an equivalent beneficial impact to that of SA, acting as biostimulant of the photosynthetic function. If this hypothesis is verified, the molecular mechanisms by which Asp enhances light energy use efficiency would be particularly valuable as it could be applied to improve PSII efficiency in crop plants.

## 2. Results

### 2.1. Leaf Water Content and Soil Water Content After Aspirin (Asp) Treatment

Twenty-four hours after the foliar spray of tomato plants with Asp, the leaf water content was estimated to be 4% higher in Asp-sprayed leaves and continued to be 4% higher after 96 h, compared to the corresponding one of water (WA)-sprayed leaves (Table 1). The volumetric soil water content in the pots of Asp-sprayed tomato plants 24 h after the spray was 7% higher and continued to be 7% higher 96 h after spraying, compared to the soil water content of the corresponding pots of WA-sprayed plants (Table 1).

### 2.2. Impact of Asp on Chlorophyll Content

The chlorophyll content did not differ significantly between WA- and Asp-sprayed leaves at the same time point, but 96 h after spraying, WA-sprayed leaves possessed 30% higher chlorophyll content compared to 24 h Asp-sprayed leaves, and 13% (ns) compared to the same time point (96 h) (Figure 1).

### 2.3. Changes in Light Energy Use Efficiency After Asp Treatment

The effective quantum yield of PSII photochemistry (Φ*_PSII_*) 24 h after spraying, at the growth light intensity (GLI, 426 μmol photons m^−2^ s^−1^), did not differ between WA- and Asp-sprayed leaves (Figure 2a), but at the high light intensity (HLI, 1000 μmol photons m^−2^ s^−1^) Φ*_PSII_* was 11% higher in Asp-sprayed leaves compared to WA-sprayed ones (Figure 2b). Ninety-six h after spraying, Φ*_PSII_* in Asp-sprayed leaves was 33% higher at the GLI (Figure 2a), and 49% higher at the HLI (Figure 2b) compared to the corresponding WA-sprayed ones.

The quantum yield of regulated non-photochemical energy loss in PSII (Φ*_NPQ_*) 24 h after the spray did not differ between WA- and Asp-sprayed leaves at both the GLI (Figure 3a) and the HLI (Figure 3b), but 96 h after spraying, Φ*_NPQ_* in Asp-sprayed leaves was 43% lower at the GLI (Figure 3a), and 22% lower at the HLI (Figure 3b), compared to the corresponding WA-sprayed ones.

The quantum yield of non-regulated energy loss in PSII (Φ*_NO_*) 24 h after spraying at the GLI did not differ between WA- and Asp-sprayed leaves (Figure 4a), but at the HLI Φ*_NO_* was 6% lower in Asp-sprayed leaves compared to WA-sprayed ones (Figure 4b). Ninety-six h after spraying, Φ*_NO_* in Asp-sprayed leaves was 7% lower at the GLI (Figure 4a) and 12% lower at the HLI (Figure 3b) compared to the corresponding WA-sprayed ones.

### 2.4. Impact of Asp on Heat Dissipation (NPQ) and on PSII Reaction Centers (qp)

Non-photochemical quenching (NPQ), which reflects heat dissipation of excitation energy 24 h after spraying, did not differ between WA- and Asp-sprayed leaves at both the GLI (Figure 5a) and the HLI (Figure 5b), but 96 h after spraying, NPQ values in Asp-sprayed leaves were 39% lower at the GLI compared to WA-sprayed ones (Figure 5a) and did not differ at the HLI (Figure 5b).

The fraction of open PSII reaction centers (q*p*), 24 h after spraying, did not differ between WA- and Asp-sprayed leaves at the GLI (Figure 6a), but at the HLI, a 12% higher fraction of reaction centers was open in Asp-sprayed leaves (Figure 6b). At 96 h after spraying, the fraction of open reaction centers in Asp-sprayed leaves was 28% higher at the GLI (Figure 6a) and 49% higher at the HLI (Figure 6b) compared to the corresponding WA-sprayed ones.

### 2.5. The Efficiency of Open PSII Reaction Centers and the Electron Transport Rate After Asp Treatment

The efficiency of the excitation energy capture by open PSII reaction centers (F*v*’/F*m*’), 24 h after spraying, did not differ between WA- and Asp-sprayed leaves at both the GLI (Figure 7a) and the HLI (Figure 7b), and also 96 h after spraying at the HLI (Figure 7b). However, at the GLI 96 h after spraying (Figure 7a), Asp-sprayed leaves showed a 4% higher efficiency of excitation energy capture by open PSII reaction centers (F*v*’/F*m*’) compared to the corresponding WA-sprayed ones.

The electron transport rate (ETR) 24 h after spraying at the GLI did not differ between WA- and Asp-sprayed leaves (Figure 8a), but at the HLI, the ETR was 11% higher in Asp-sprayed leaves compared to WA-sprayed ones (Figure 8b). Ninety-six hours after spraying, the ETR in Asp-sprayed leaves was 33% higher at the GLI (Figure 8a) and 49% higher at the HLI (Figure 8b) compared to the corresponding ETR in WA-sprayed ones.

### 2.6. The Excitation Pressure and the Excess Excitation Energy at PSII After Asp Treatment

The excitation pressure at PSII (1 − q*L*), 24 h after spraying, did not differ between WA- and Asp-sprayed leaves at the GLI (Figure 9a), but at the HLI, the excitation pressure (1 − q*L*), in Asp-sprayed leaves was 8% lower compared to the corresponding WA-sprayed ones (Figure 9b). At 96 h after spraying, excitation pressure (1 − q*L*) in Asp-sprayed leaves was 26% lower at the GLI (Figure 9a) and 21% lower at the HLI (Figure 9b), compared to the corresponding excitation pressure in WA-sprayed ones.

The excess excitation energy (EXC), 24 h after spraying, did not differ between WA- and Asp-sprayed leaves at the GLI (Figure 10a), but at the HLI, the EXC in Asp-sprayed leaves was 13% lower compared to the corresponding WA-sprayed ones (Figure 10b). At 96 h after spraying, EXC in Asp-sprayed leaves was 40% lower at the GLI (Figure 9a) and 35% lower at the HLI (Figure 9b) compared to the corresponding WA-sprayed ones.

### 2.7. Correlation of the Excitation Pressure in PSII with the Excess Excitation Energy at PSII

The relationship between the excess excitation energy (EXC) and the excitation pressure in PSII (1 − q*L*) is depicted in Figure 11a,b. The graph is based on the data from the Figure 9a,b and Figure 10a,b. The excitation pressure at PSII was significantly correlated with the amount of the excess excitation energy at the GLI (*p* < 0.001, *R*^2^ = 0.9805) (Figure 11a) and at the HLI (*p* < 0.001, *R*^2^ = 0.9930) (Figure 11b).

### 2.8. Changes in Reactive Oxygen Species Generation After Asp Treatment.

A decrease in the generation of reactive oxygen species (ROS) was noticed in leaflets sprayed with Asp (Figure 12b) 24 h after spraying, as compared to the WA-sprayed leaflets (Figure 12a). The higher ROS production, localized mainly in the leaf veins of water-sprayed leaflets, was detectable as green fluorescence (Figure 12a), while in the leaf veins of Asp-sprayed leaflets, a lower ROS production was observed (Figure 12b).

### 2.9. Chloroplast Ultrastructural Changes After Asp Treatment

The chloroplasts of leaves sprayed both with water and Asp exhibited well-developed grana thylakoids and frets (Figure 13). However, in WA-sprayed leaves, the chloroplasts were devoid of starch and appeared to be accompanied by numerous peroxisomes, which included prominent electron-dense crystals (Figure 13a,b). On the contrary, in Asp-sprayed leaves, the chloroplasts contained grains of starch, while peroxisomes were not encountered at chloroplast proximity (Figure 13c,d).

## 3. Discussion

Molecular mechanisms induced by SA that can enhance the synthesis of osmolytes, like proline, are capable of maintaining a high leaf water potential [17,18,19,20,21,54,55,56,57]. In our experiments, Asp-sprayed tomato leaflets 24 h and 96 h after spraying possessed an increased leaf water content compared to WA-sprayed ones (Table 1), suggesting the induction of osmolyte synthesis. In common beans under cold stress, ASA application, increased total soluble sugars and proline accumulation [45]. Exogenous application of Asp, that acts as an osmo-regulator, can be a favorable tool to increase drought tolerance which is a growing agricultural issue [16,54,56]. The Asp-sprayed plants in addition to increased leaf water content could maintain a superior amount of soil water content under both 24 h and 96 h measurements, compared to WA-sprayed ones (Table 1). This can be explained by a lower transpiration rate of the Asp-sprayed plants. Accordingly, a reduced fraction of closed PSII reaction centers (1 − q*L*), has been shown to be linearly correlated with a decreased stomatal conductance [57,58]. Thus, the decreased number of closed PSII reaction centers (1 − q*L*) in Asp treated leaves (Figure 9a,b) is correlated with a reduced stomatal opening, suggesting a reduced transpiration.

Chlorophylls are the main pigments in the light-harvesting complexes (antenna) of photosystem I (PSI) and photosystem II (PSII). Chlorophylls can absorb the light energy that is transferred to the reaction centers where charge separation occurs, and electron transport is initiated [4,59]. If the absorbed light energy exceeds the amount that can be used in photochemistry and cannot be dissipated by the photoprotective mechanism of non-photochemical quenching (NPQ), an increased ROS generation occurs that results in photo-oxidative stress [37,39,59,60,61,62,63]. As an outcome photodamage of PSII occurs through triplet excited state chlorophylls (^3^Chl*), that result in singlet oxygen (^1^O_2_) formation [64,65,66]. Chlorophyll content in both WA-sprayed and Asp-sprayed tomato leaflets tends to increase with time after spraying, but it remains always higher in WA-sprayed leaves at the same time point. SA is known to act as a signaling molecule triggering chlorophyll catabolic genes [67]. It seems that Asp exerts a similar physiological role with SA in its influence on chlorophyll molecules. The increase or decrease of chlorophyll content by SA are connected to the plant species, the genotype, the concentration of SA, the duration of exposure and the environmental conditions [4,9,14,68,69,70,71,72,73].

A lower chlorophyll content has been ascribed to smaller antenna size and less light energy absorption, which results in lower ROS generation [74,75,76,77,78]. Downscaling of the light-harvesting capacity can avoid photo-oxidative stress [75,79,80]. Thus, lower leaf chlorophyll content decreases sunlight absorption and enhances photosynthetic function by diminishing photo-oxidative stress, especially in high light environments [74,75,76,77,78,81]. Improved photosynthetic efficiency can be achieved via better allocation of absorbed light energy which also reduces photo-oxidative stress [82]. It should be mentioned that both leaf age and light intensity play crucial roles in determining PSII responses [83].

The effective quantum yield of PSII photochemistry (Φ*_PSII_*) was higher in Asp-sprayed leaves 96 h after the spray, at both the GLI (11%) (Figure 2a) and the HLI (49%) (Figure 2b), while at the HLI, Φ*_PSII_* was higher (11%) even at 24 h after the spray (Figure 2b). The increased values of Φ*_PSII_* in Asp-sprayed leaves (Figure 2a,b) were mainly due to the increased number of open PSII RCs (q*p*) (Figure 6a,b), since the efficiency of the open PSII RCs (F*v*’/F*m*’), almost did not differ between WA-sprayed and Asp-sprayed leaves (Figure 7a,b). The quantum yield of regulated non-photochemical energy loss in PSII (Φ*_NPQ_*), 96 h after spraying decreased in Asp-sprayed leaves at both the GLI (43%) (Figure 3a) and the HLI (22%) (Figure 3b), compared to the corresponding WA-sprayed ones.

The quantum yield of non-regulated energy loss in PSII (Φ*_NO_*), verifying the better light energy use efficiency in the Asp-sprayed leaves also decreased 96 h after spraying at both the GLI (7%) (Figure 4a) and the HLI (12%) (Figure 4b), compared to the corresponding Φ*_NO_* values in WA-sprayed ones. Decreased Φ*_NO_* values symbolize decreased triplet chlorophyll state (^3^Chl*) generation that has, as a consequence, a lower singlet oxygen (^1^O_2_) formation [64,84,85]. The very reactive ^1^O_2_ is produced by the reaction of the excited ^3^Chl* with molecular O_2_ [39,64]. Therefore, the possibility of ^1^O_2_ development can be calculated by Φ*_NO_* [85,86,87,88,89]. Thus, reduced Φ*_NO_* values unveils a lesser ^1^O_2_ generation and a better ability of the plant to safeguard itself versus excess light energy [59,86,87,88,89,90], suggesting an antioxidant photoprotection offered by Asp. These data are validated by the lower ROS production detected in the leaf veins of Asp sprayed leaflets (Figure 12b). ROS creation in the light reactions of photosynthesis spreads throughout the leaf veins to act as a long-distance signalling molecule [84,91,92,93,94,95] to initiate defense responses during abiotic and biotic stress conditions [39,96,97]. Different ROS signalling pathways are triggered by the redox state of Q_A_, that involves a mechanism of plant acclimation which regulates photosynthetic gene expression, achieving the appropriate response to various environmental stresses [93,98,99,100].

The heat dissipation of excitation energy for photoprotection (NPQ) decreased (39%) 96 h after spraying in Asp-sprayed leaves at the GLI (Figure 5a), while NPQ did not differ at the HLI (Figure 5b) between WA- and Asp-sprayed leaves. Also, NPQ did not differ between WA- and Asp-sprayed leaves 24 h after spraying at both the GLI (Figure 5a) and the HLI (Figure 5b). However, despite a non-difference in NPQ between WA- and Asp-sprayed leaves or a decreased NPQ, 96 h after Asp-spraying, the fraction of open PSII reaction centers increased in Asp-sprayed leaves, compared to WA-sprayed ones, especially at the HLI (Figure 6b). Thus, the NPQ mechanism in Asp sprayed leaves was more than sufficient since it increased the amount of open PSII reaction centers [101,102]. Therefore, Asp acted as an antioxidant and offered a better photoprotection, keeping the redox state of the PQ pool (q*p*) more oxidized (Figure 6a,b). The redox state of the PQ pool is considered fundamental for retrograde signaling [103,104,105] and has significant importance for antioxidant defense and signaling [106].

Comparing the impact of SA with that of Asp on the light energy use efficiency in PSII, we noticed that Φ*_PSII_* in basil plants 96h after the spray with 1mM SA increased by 40% at 900 μmol photons m^−2^ s^−1^ [15], while Φ*_PSII_* in Asp-sprayed tomato leaves was 49% higher at 1000 μmol photons m^−2^ s^−1^, compared to the corresponding WA-sprayed ones. Yet, the fraction of open PSII reaction centers (q*p*) at 900 μmol photons m^−2^ s^−1^, 96h after spraying with 1mM SA, increased by 52% [15], while at 1000 μmol photons m^−2^ s^−1^, 96 h after Asp-sprayed, was 49% higher compared to the corresponding WA-sprayed ones. The lower increase of Φ*_PSII_* in basil plants, despite a higher increase of q*p,* was due to the 9% decreased efficiency of the excitation energy capture by open PSII reaction centers (F*v*’/F*m*’), 96 h after the spray with SA [15], while in Asp-sprayed tomato leaves 96h after spraying there was no difference in F*v*’/F*m*’, compared to the corresponding WA-sprayed leaves. Thus, it seems that Asp could be considered a better biostimulant of the photosynthetic function compared to SA, with whom it shares similar physiological characteristics. However, it is important to notice that the impact of all biostimulants may differ with the plant species, the concentration used, and the exposure time, but also with the way of application [4,15,22].

In WA-sprayed leaves, the chloroplasts were accompanied by numerous peroxisomes containing prominent electron-dense crystals (Figure 13a,b), indicating the involvement of a high photorespiration process functioning as a major alternative sink [107]. Photorespiration acts as a safety valve for photoprotection [107], safeguarding photosynthesis, preventing the over-reduction of the electron transport chain, and restricting ROS production [108,109,110]. However, this process reduces overall photosynthetic yield [110], and net carbon fixation [111,112], characterized as an energy-expensive process [113]. Reducing photorespiration has been shown to enhance photosynthetic capacity and increase potato tuber mass, making it a promising target for improving photosynthetic efficiency [114]. In contrast to WA-sprayed leaves, in Asp-sprayed leaves there was no need for photoprotection, because it was offered by Asp, and was proved to be more effective than the photorespiration process in WA-sprayed leaves, based on the lower ROS generation in Asp sprayed leaves (Figure 12b). In addition, the increased starch accumulation in Asp-sprayed leaves suggests the absence of cyclic electron flow [115] that is also employed for preventing over-reduction of the plastoquinone pool and ROS production, but results in a reduced photochemical efficiency. These chloroplast ultrastructural data confirm the conclusion that spraying with Asp provided antioxidant protection and photoprotection, especially under excess light intensity. In addition Asp-sprayed leaves possessed a better light energy use efficiency compared to WA-sprayed leaves, with an increased photochemistry. The photoprotective effect provided by Asp, especially under excess light illumination, is recognized also by the reduced excess excitation energy that resulted to a decreased excitation pressure on tomato PSII. In agreement to this, a strong correlation between the excitation pressure and the amount of the excess excitation energy, at both the GLI (Figure 11a), and the HLI (Figure 11b) was revealed in tomato PSII function.

## 4. Materials and Methods

### 4.1. Plant Material and Growth Conditions

Tomato (*Solanum lycopersicum* L. cv Gardeners Ecstasy) plants grew in a greenhouse with a 14-h photoperiod, 24 ± 1/20 ± 1 °C (day/night) temperature, relative humidity 65 ± 5/75 ± 5% (day/night), and photosynthetic photon flux density (PPFD) of 400 ± 10 μmol quanta m^−2^ s^−1^.

### 4.2. Soil and Leaf Water Water Content

The ProCheck device connected to a 5TE sensor (Decagon Devices, Pullman, WA, USA) was used to measure soil volumetric water content (SWC) that was expressed in m^3^ m^−3^. The leaf water content of tomato leaves was assessed using the electronic moisture balance (MOC120H, Shimadzu, Tokyo, Japan) that was expressed as %.

### 4.3. Asp Treatment

Asp tablets were purchased from the local pharmacy. One Aspirin^®^ (Bayer Hellas, Maroussi, Athens, Greece) tablet of 100 mg was dissolved in 100 mL distilled water, and each tomato plant was foliar sprayed with 10 mL solution. Control plants were sprayed with 10 mL of distilled water. Five plants and two independent replicates were used in each treatment.

### 4.4. Chlorophyll Content

The relative chlorophyll content in tomato leaflets was measured photometrically with a portable Chlorophyll Content Meter (Model Cl-01, Hansatech Instruments Ltd., Norfolk, UK) and expressed in relative units [4,116].

### 4.5. Chlorophyll Fluorescence Analysis

Chlorophyll fluorescence measurements were performed with the Imaging-PAM Fluorometer M-Series MINI-Version (Heinz Walz GmbH, Effeltrich, Germany), as described in detail previously [117]. All leaves were dark-adapted for 20 min before measuring the minimum (F*o*) and the maximum (F*m*) chlorophyll *a* fluorescence in the dark. The maximum chlorophyll *a* fluorescence in the light (F*m*’), was obtained with saturating pulses (SPs) every 20 s for 5 min after application of the actinic light (AL), while the minimum chlorophyll *a* fluorescence in the light (F*o*’) was computed as F*o*’ = F*o*/(F*v*/F*m* + F*o*/F*m*’) [118]. Steady-state photosynthesis (F*s*) was measured after 5 min of illumination time with the AL of 426 μmol photons m^−2^ s^−1^, corresponding to the growth light intensity (GLI), and with 1000 μmol photons m^−2^ s^−1^ AL, corresponding to a high light intensity (HLI). By using Win V2.41a software (Heinz Walz GmbH, Effeltrich, Germany), we estimated the chlorophyll fluorescence parameters, described in Appendix A.

### 4.6. Imaging of Reactive Oxygen Species

Reactive oxygen species (ROS) imaging was performed by incubating WA-sprayed and Asp-sprayed tomato leaflets in 25 μM 2′,7′-dichlorofluorescein diacetate (DCF-DA, Sigma Aldrich, Chemie GmbH, Schnelldorf, Germany) for 30 min in the dark [97,119]. Then ROS specific fluorescence was observed with a Zeiss AxioImager Z2 epi-fluorescence microscope (Carl Zeiss MicroImaging GmbH, Göttingen, Germany) equipped with an AxioCam MRc5 digital camera (Carl Zeiss MicroImaging GmbH, Göttingen, Germany) [119].

### 4.7. Transmission Electron Microscope Observations

WA-sprayed and Asp-sprayed tomato leaves were cut in pieces of ~2 × 2 mm^2^ with razor blades and immediately fixed in 3% glutaraldehyde in 50 mM sodium cacodylate buffer (pH 7) for 4 h at room temperature, as described before [16]. After three rinses in the same buffer, the samples were post-fixed at 4 °C overnight in 1% osmium tetroxide in the same buffer. After rinsing, the samples were dehydrated in acetone series, treated 2 × 20 min with propylene oxide at 4 °C, and lastly, embedded in Spurr’s resin. Ultrathin sections (~70 nm) were double stained with uranyl acetate and lead citrate and examined with a JEOL JEM 1011 (JEOL Tokyo, Japan) transmission electron microscope at 80 kV. Digital electron micrographs were acquired with a GATAN 500 camera (Gatan, Pleasanton, CA, USA).

### 4.8. Statistics

Statistically significant differences were evaluated by two-way ANOVA for each parameter with treatment (Water or Asp) and time (24 or 96 h) as factors, followed by Tukey’s post hoc test. The data were checked for normality and homogeneity of variance with Shapiro–Wilk test and Levene’s test, respectively. All statistical analyses were performed with R software (version 4.3.1, R Core Team, 2023). Values were considered significantly different at *p* < 0.05.

## 5. Conclusions

Foliar Asp-spraying of tomato leaves increased photochemistry (Φ*_PSII_*), ETR, and the number of open reaction centers (q*p*), while it resulted in less excess excitation energy and in a lower ^1^O_2_ generation, suggesting antioxidant protection. In addition, it resulted in a decreased stomatal opening that reduced transpiration and yet resulted in a significantly lower quantum yield for heat dissipation (Φ*_NPQ_*), indicating the photoprotective effect provided by Asp, especially under excess light illumination. It is concluded that Asp shares similar physiological characteristics with SA, having an equivalent beneficial impact to that of SA by acting as a biostimulant of the photosynthetic function for an enhanced crop yield.

## Figures and Tables

**Figure 1 ijms-26-01368-f001:**
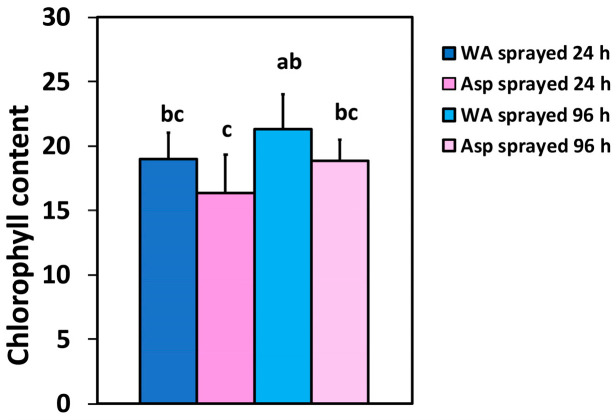
The chlorophyll content of water (WA)-sprayed and Aspirin (Asp)-sprayed leaves 24- and 96-h after the spray, expressed in relative units (*n* = 10 ± SD). Significant differences are shown by different lower-case letters (*p* < 0.05).

**Figure 2 ijms-26-01368-f002:**
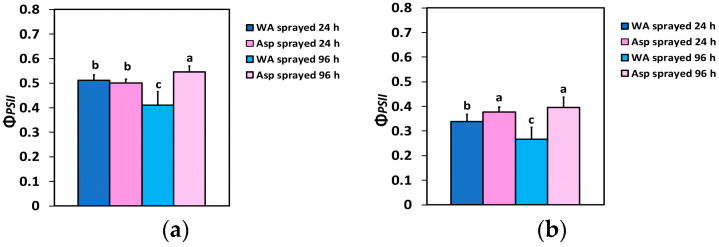
The effective quantum yield of PSII photochemistry (Φ*_PSII_*) at the growth light intensity (GLI) (**a**) and at the high light intensity (HLI) (**b**) of WA-sprayed and Asp-sprayed leaves 24- and 96-h after the spray (*n* = 6 ± SD). Significant differences are shown by different lower-case letters (*p* < 0.05).

**Figure 3 ijms-26-01368-f003:**
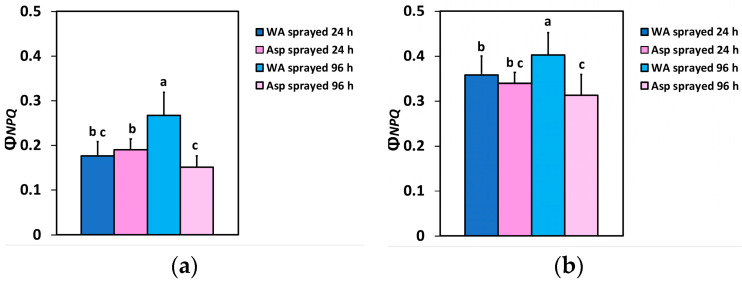
The quantum yield of regulated non-photochemical energy loss in PSII (Φ*_NPQ_*) at the GLI (**a**) and at the HLI (**b**) of WA-sprayed and Asp-sprayed leaves 24- and 96-h after the spray (*n* = 6 ± SD). Significant differences are shown by different lower-case letters (*p* < 0.05).

**Figure 4 ijms-26-01368-f004:**
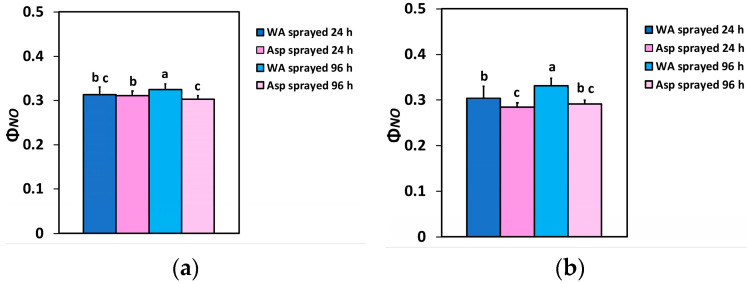
The quantum yield of non-regulated energy loss in PSII (Φ*_NO_*) at the GLI (**a**) and at the HLI (**b**) of WA-sprayed and Asp-sprayed leaves 24- and 96-h after the spray (*n* = 6 ± SD). Significant differences are shown by different lower-case letters (*p* < 0.05).

**Figure 5 ijms-26-01368-f005:**
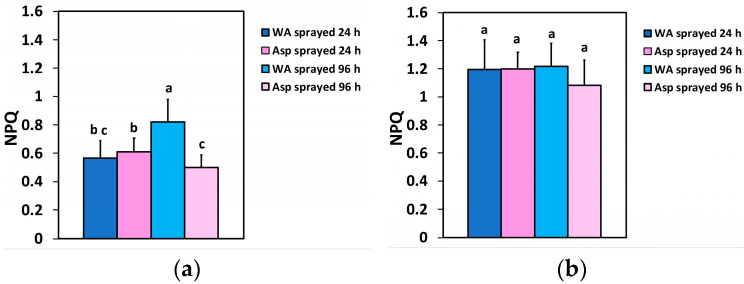
The non-photochemical quenching (NPQ), at the GLI (**a**) and at the HLI (**b**) of WA-sprayed and Asp-sprayed leaves 24- and 96-h after the spray (*n* = 6 ± SD). Significant differences are shown by different lower-case letters (*p* < 0.05).

**Figure 6 ijms-26-01368-f006:**
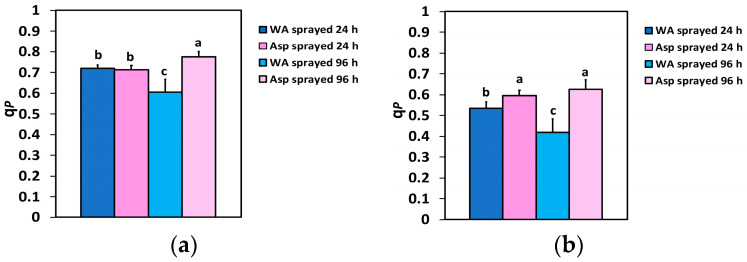
The fraction of open PSII reaction centers (RCs) (q*p*), at the GLI (**a**) and at the HLI (**b**) of WA-sprayed and Asp-sprayed leaves 24- and 96-h after the spray (*n* = 6 ± SD). Significant differences are shown by different lower-case letters (*p* < 0.05).

**Figure 7 ijms-26-01368-f007:**
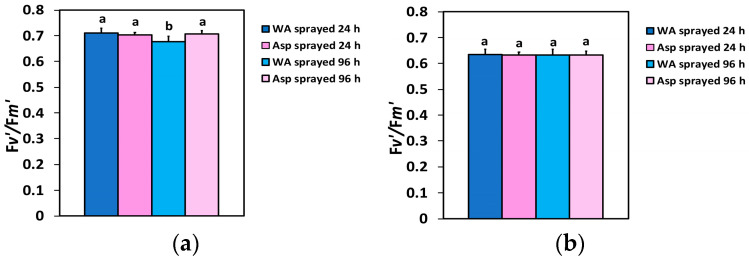
The efficiency of the open PSII RCs (F*v*’/F*m*’) at the GLI (**a**) and at the HLI (**b**) of WA-sprayed and Asp-sprayed leaves 24- and 96-h after the spray (*n* = 6 ± SD). Significant differences are shown by different lower-case letters (*p* < 0.05).

**Figure 8 ijms-26-01368-f008:**
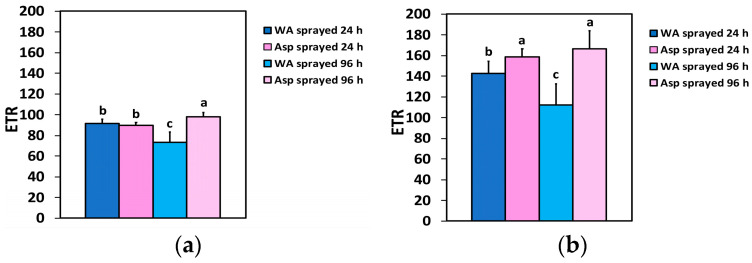
The electron transport rate (ETR) at the GLI (**a**) and at the HLI (**b**) of WA-sprayed and Asp-sprayed leaves 24- and 96-h after the spray (*n* = 6 ± SD). Significant differences are shown by different lower-case letters (*p* < 0.05).

**Figure 9 ijms-26-01368-f009:**
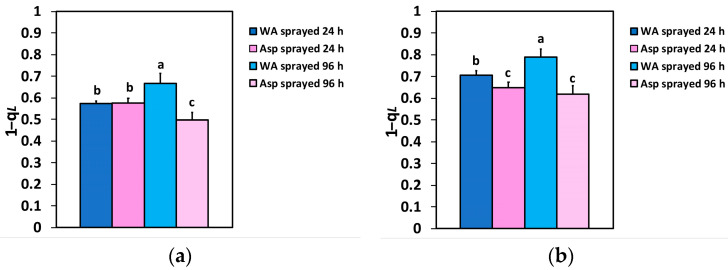
The excitation pressure at PSII (1 − q*L*), measured at the GLI (**a**) and at the HLI (**b**) of WA-sprayed and Asp-sprayed leaves 24- and 96-h after the spray (*n* = 6 ± SD). Significant differences are shown by different lower-case letters (*p* < 0.05).

**Figure 10 ijms-26-01368-f010:**
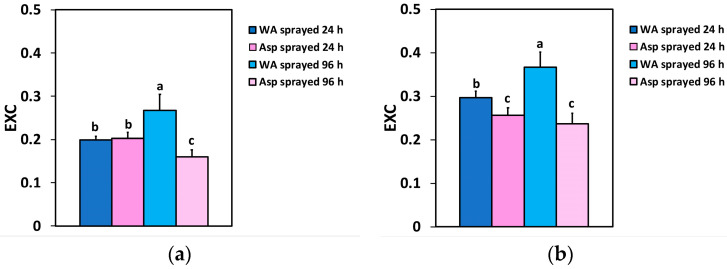
The excess excitation energy at PSII (EXC), at the GLI (**a**) and at the HLI (**b**) of WA- sprayed and Asp-sprayed leaves 24- and 96-h after the spray (*n* = 6 ± SD). Significant differences are shown by different lower-case letters (*p* < 0.05).

**Figure 11 ijms-26-01368-f011:**
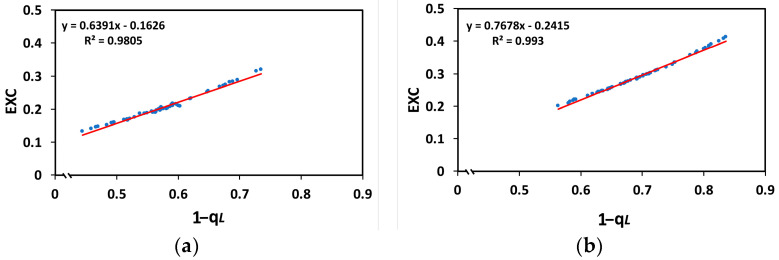
The relationship between the excess excitation energy (EXC) and the excitation pressure at PSII (1 − q*L*) at the GLI (**a**) and at the HLI (**b**) of WA- sprayed and Asp-sprayed leaves 24- and 96-h after the spray (based on the data of Figure 9a,b and Figure 10a,b). Each blue dot represents the paired measurement of the variables, while the red line is the regression line that shows the relationship between the two variables.

**Figure 12 ijms-26-01368-f012:**
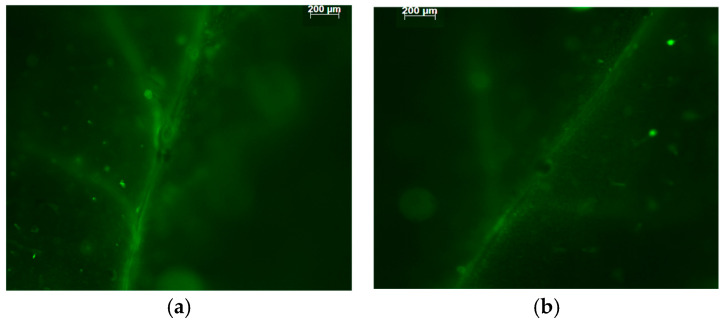
ROS production in tomato leaflets of WA-sprayed (**a**) and Asp-sprayed **(b**) leaves 24-h after the spray. The light green color indicates ROS generation. Scale bar, 200 μm.

**Figure 13 ijms-26-01368-f013:**
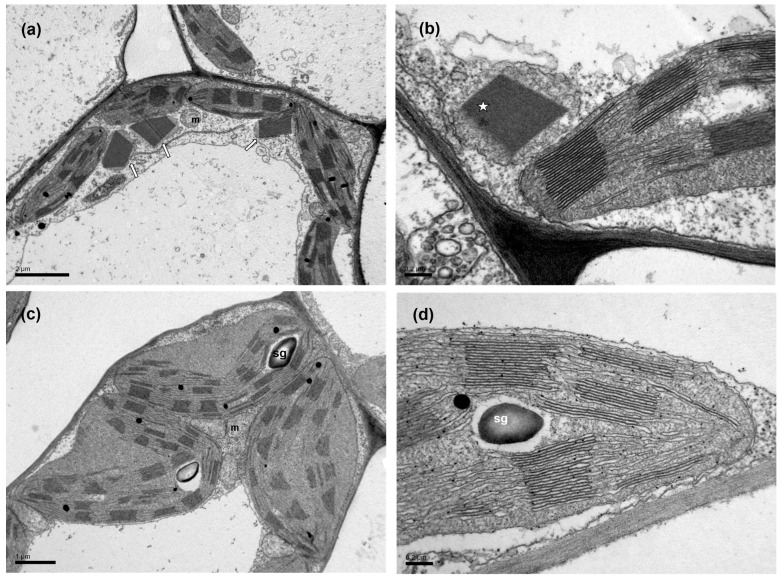
Transmission electron micrographs of mesophyll cells of tomato leaves sprayed with water (**a**,**b**) or with Asp (**c**,**d**). Note the peroxisomes (arrows in (**a**)), which include electron-dense crystals (asterisk in (**b**)) in cells of WA-sprayed leaves. Starch grains (sg) can be observed in chloroplasts of Asp-sprayed leaves (**c**,**d**) but not in those of WA-sprayed leaves. m: mitochondrion. Scale bars as indicated on the micrographs.

**Table 1 ijms-26-01368-t001:** The leaf water content of water (WA)-sprayed and Aspirin (Asp)-sprayed tomatoes (*n* = 10 ± SD) and the soil water content of their respective pots (*n* = 6 ± SD).

Parameter	WA-Sprayed 24 h	Asp-Sprayed 24 h	WA-Sprayed 96 h	Asp-Sprayed 96 h
Leaf Water Content ^1^	86.36 ± 0.012	90.10 ± 0.014	85.51 ± 0.011	88.86 ± 0.012
Soil Water Content ^2^	0.482 ± 0.032	0.514 ± 0.028	0.453 ± 0.052	0.484 ± 0.015

^1^ expressed %; ^2^ expressed in m^3^ m^−3^.

## Data Availability

The data presented in this study are available in this article.

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
