# Peer review of "Aspirin Foliar Spray-Induced Changes in Light Energy Use Efficiency, Chloroplast Ultrastructure, and ROS Generation in Tomato"

_ijms, 2025, doi:10.3390/ijms26031368_

Round 1
Reviewer 1 Report
Comments and Suggestions for Authors
This article manuscript investigates how Aspirin Foliar Spray have some effect on Light Energy Efficiency, Chloroplast Ultrastructure and ROS Generation in Tomato. The research topic is of great interest, also has potential to publish after revision.
1. The chemical and physiological characteristic similarities between ASA and SA. Why not use SA to compare difference in ight Energy Efficiency, Chloroplast Ultrastructure and ROS Generation
2. The authors should include a thorough explanation in the manuscript as to why they studied only two intervals (i.e., 24 and 96 h of aspirin treatment).
3. In Figure 1, The Y-axis should be chlorophyll content
Author Response
This article manuscript investigates how Aspirin Foliar Spray have some effect on Light Energy Efficiency, Chloroplast Ultrastructure and ROS Generation in Tomato. The research topic is of great interest, also has potential to publish after revision.
Comment 1: The chemical and physiological characteristic similarities between ASA and SA. Why not use SA to compare difference in light Energy Efficiency, Chloroplast Ultrastructure and ROS Generation
Answer: We have previously examined the effects of salicylic acid on photosystem II function (Ref. 4,14,15 and 16) and thus in this manuscript we used Aspirin for comparison.
Comment 2: The authors should include a thorough explanation in the manuscript as to why they studied only two intervals (i.e., 24 and 96 h of aspirin treatment).
Answer: On the basis of our previous results (Ref. 4,14,15 and 16) and those preliminary obtained in this study we decided to use these two-time intervals that could be used also for comparison with our previous results.
Comment 3: In Figure 1, The Y-axis should be chlorophyll content
Answer: Yes, we inserted in Figure 1, chlorophyll content in the Y-axis.
Reviewer 2 Report
Comments and Suggestions for Authors
This manuscript presents a thorough investigation into the role of aspirin (Asp) on the photosynthesis of tomato plants. The study meticulously examines the physical indicators, which are scientific and reasonable. The comprehensive analysis provides valuable insights into the new biostimulant option for plant growth regulation. The article exhibits clear logic. However, due to the previous studies on the effects of salicylic acid analogs on plants, the overall innovation of this study is somewhat lacking. Other comments are as follows:
1. This study only set a single concentration of Asp. It is better to set a concentration gradient xperiment of aspirin treatment to comprehensively understand the relationship between the aspirin dose and tomato responses.
2. It is recommeded to conduct multivariate statistical methods such as principal component analysis and correlation analysis to comprehensively revealing the internal connections between various physiological indicators and the comprehensive effects of aspirin treatment on tomato photosynthesis.
3. For the figures, the description of some figures are not detailed enough. For example, the units of the coordinate axes are not clearly defined in Figure 1.
Comments on the Quality of English LanguageThe english writing should be improved.
Author Response
This manuscript presents a thorough investigation into the role of aspirin (Asp) on the photosynthesis of tomato plants. The study meticulously examines the physical indicators, which are scientific and reasonable. The comprehensive analysis provides valuable insights into the new biostimulant option for plant growth regulation. The article exhibits clear logic. However, due to the previous studies on the effects of salicylic acid analogs on plants, the overall innovation of this study is somewhat lacking. Other comments are as follows:
Comment 1: This study only set a single concentration of Asp. It is better to set a concentration gradient experiment of aspirin treatment to comprehensively understand the relationship between the aspirin dose and tomato responses.
Answer: We used only a single concentration of Asp on the basis of preliminary experiments that revealed that Asp at higher concentration could not be completely dissolved in aqueous solution and we did not want to use an organic solvent to increase solubility. Yet, at lower Asp concentration the effect was less obvious on light energy use efficiency of tomato leaves. Organic solvents (e.g. ethanol) have a negative impact on PSII function.
Comment 2: It is recommeded to conduct multivariate statistical methods such as principal component analysis and correlation analysis to comprehensively revealing the internal connections between various physiological indicators and the comprehensive effects of aspirin treatment on tomato photosynthesis.
Answer: PCA analysis is usually used on large datasets when the direct interpretation of the interrelated variables is difficult, or to uncover hidden patterns and relationships in the data. In our experiment, we do not have datasets with that level of complexity (only one plant species, and one treatment with 2 time points) and at the same time the relationships between the measured parameters are well known. However, we performed a correlation analysis that revealed the close connection between the amount of excess excitation energy and the excitation pressure on tomato plants (New Figure 11). A relative sentence was added in the Discussion (Lines 629-633).
Comment 3: For the figures, the description of some figures are not detailed enough. For example, the units of the coordinate axes are not clearly defined in Figure 1.
Answer: The units of Figure 1 were defined in the legend of Figure 1.
Comments on the Quality of English Language
The english writing should be improved.
Answer: The English language was checked and improved.
Reviewer 3 Report
Comments and Suggestions for Authors
The manuscript entitled “Aspirin Foliar Spray-Induced Changes in Light Energy Use Efficiency, Chloroplast Ultrastructure and ROS Generation in Tomato”
This study investigates the effects of foliar aspirin (Asp) spray on tomato plants, comparing it to salicylic acid (SA) due to their similar physiological roles. The results show that Asp enhances photosystem II (PSII) efficiency, reduces reactive oxygen species (ROS) formation, and improves the redox state of plastoquinone (PQ). Asp also decreases stomatal opening, reducing transpiration, and offers photoprotection under excess light. Additionally, it helps maintain chloroplast integrity, promoting better photosynthetic performance and potentially enhancing crop yield.
The manuscript is well-written, with a compelling topic that fits the journal's scope. The experiments are well-designed and thoroughly conducted, and the conclusions are convincing. However, I recommend acceptance for publication, after addressing below minor comments:
1. The author should compare aspirin with other biostimulants to assess its relative efficacy and potential for broader agricultural use.
2. The author should avoid providing the full explanation of ASP in each instance where it is mentioned. This information should be omitted such as Page number 3, line 98.
3. In the Materials and Methods section, the author should include detailed information regarding the purchase of aspirin.
4. Please ensure consistent spacing between the text throughout the manuscript. For example, on page 8, line 209, there is no space between "Figure 12a,b," while on line 211, "Figure 12c,d" has proper spacing. This inconsistency should be corrected.
5. The author should review all references and ensure that page numbers are included, and the year is bolded for the following references: 12, 13, 18, 48, 34, 35, 53, 78, and 80.
Author Response
The manuscript entitled “Aspirin Foliar Spray-Induced Changes in Light Energy Use Efficiency, Chloroplast Ultrastructure and ROS Generation in Tomato”
This study investigates the effects of foliar aspirin (Asp) spray on tomato plants, comparing it to salicylic acid (SA) due to their similar physiological roles. The results show that Asp enhances photosystem II (PSII) efficiency, reduces reactive oxygen species (ROS) formation, and improves the redox state of plastoquinone (PQ). Asp also decreases stomatal opening, reducing transpiration, and offers photoprotection under excess light. Additionally, it helps maintain chloroplast integrity, promoting better photosynthetic performance and potentially enhancing crop yield.
The manuscript is well-written, with a compelling topic that fits the journal's scope. The experiments are well-designed and thoroughly conducted, and the conclusions are convincing. However, I recommend acceptance for publication, after addressing below minor comments:
The author should compare aspirin with other biostimulants to assess its relative efficacy and potential for broader agricultural use.
Answer: We added a new paragraph comparing the impact of aspirin with that of SA on the light energy use efficiency in PSII, and concluded that Asp could be considered a better biostimulant of the photosynthetic function compared to SA (lines 594-609).
- The author should avoid providing the full explanation of ASP in each instance where it is mentioned. This information should be omitted such as Page number 3, line 98.
Answer: Full explanation of ASP was omitted after the first mention.
- In the Materials and Methods section, the author should include detailed information regarding the purchase of aspirin.
Answer: Information regarding the purchase of aspirin was included in Materials and Methods section.
- Please ensure consistent spacing between the text throughout the manuscript. For example, on page 8, line 209, there is no space between "Figure 12a,b," while on line 211, "Figure 12c,d" has proper spacing. This inconsistency should be corrected.
Answer: The same spacing was applied to same cases.
- The author should review all references and ensure that page numbers are included, and the year is bolded for the following references: 12, 13, 18, 48, 34, 35, 53, 78, and 80.
Answer: The reference list was checked and corrected.